# Nitrogen Dioxide Sensing Using Multilayer Structure of Reduced Graphene Oxide and α-Fe_2_O_3_

**DOI:** 10.3390/s21031011

**Published:** 2021-02-02

**Authors:** Tadeusz Pisarkiewicz, Wojciech Maziarz, Artur Małolepszy, Leszek Stobiński, Dagmara Agnieszka Michoń, Aleksandra Szkudlarek, Marcin Pisarek, Jarosław Kanak, Artur Rydosz

**Affiliations:** 1Institute of Electronics, AGH University of Science and Technology, Al. Mickiewicza 30, 30-059 Kraków, Poland; pisar@agh.edu.pl (T.P.); maziarz@agh.edu.pl (W.M.); dagmaramichon@agh.edu.pl (D.A.M.); kanak@ichf.edu.pl (J.K.); 2Faculty of Chemical and Process Engineering, Warsaw University of Technology, Waryńskiego 1, 00-645 Warsaw, Poland; artur.malolepszy@pw.edu.pl (A.M.); lstob50@hotmail.com (L.S.); 3Academic Centre for Materials and Nanotechnology, AGH University of Science and Technology, Al. Mickiewicza 30, 30-059 Kraków, Poland; aleszku@agh.edu.pl; 4Institute of Physical Chemistry, Polish Academy of Sciences, Kasprzaka 44/52, 01-224 Warsaw, Poland; mpisarek@agh.edu.pl

**Keywords:** graphen oxide, reduced graphene oxide, α-Fe_2_O_3_/rGO multilayer, NO_2_ sensing

## Abstract

Multilayers consisting of graphene oxide (GO) and α-Fe_2_O_3_ thin layers were deposited on the ceramic substrates by the spray LbL (layer by layer) coating technique. Graphene oxide was prepared from graphite using the modified Hummers method. Obtained GO flakes reached up to 6 nanometers in thickness and 10 micrometers in lateral size. Iron oxide Fe_2_O_3_ was obtained by the wet chemical method from FeCl_3_ and NH_4_OH solution. Manufactured samples were deposited as 3 LbL (GO and Fe_2_O_3_ layers deposited sequentially) and 6 LbL structures with GO as a bottom layer. Electrical measurements show the decrease of multilayer resistance after the introduction of the oxidizing NO_2_ gas to the ambient air atmosphere. The concentration of NO_2_ was changed from 1 ppm to 20 ppm. The samples changed their resistance even at temperatures close to room temperature, however, the sensitivity increased with temperature. Fe_2_O_3_ is known as an n-type semiconductor, but the rGO/Fe_2_O_3_ hybrid structure behaved similarly to rGO, which is p-type. Both chemisorbed O_2_ and NO_2_ act as electron traps decreasing the concentration of electrons and increasing the effective multilayer conductivity. An explanation of the observed variations of multilayer structure resistance also the possibility of heterojunctions formation was taken into account.

## 1. Introduction

Nitrogen dioxide is a typical air pollutant, being a component of motor vehicle exhaust gas and is also generated in the house and industrial combustion processes. High-temperature combustion leads to an increase in NOx emissions. The early applications of NOx detectors for diesel engine exhausts utilized dense zirconia-based electrolytes. It is shown in more recent investigations of nitrogen oxide sensors that good results can be obtained by integrating the sensor with NOx trap materials [1] or by the use of LaSrMnO_3_ type perovskite electrodes [2]. However, the sensors connected with vehicles exhaust work at elevated temperatures, frequently exceeding 300 °C. Exposures to NO_2_ are also dangerous for the respiratory system; the presence of nitrogen dioxide in the atmosphere is a cause of photochemical smog and acid rains. Manufacturing of inexpensive NO_2_ sensors working at low temperatures is therefore of high importance.

One of the materials recently investigated in gas sensors technology is graphene and structures based on graphene. There are many attempts to increase the sensitivity of graphene-based sensors in view of interaction with the ambient gas atmosphere, enabling practical applications. Pure monolayer graphene with a large surface area and a high conductivity is a good candidate for high-performance gas sensors. However, its production methods as epitaxial growth or chemical vapor deposition, are known as cost-prohibitive with small-scale yield. Modified graphene materials, i.e., graphene oxide (GO) or its reduced form (rGO), can be much easier manufactured, and their specific properties caused by lattice defects connected with, e.g., oxygen functional groups enhance absorption of ambient gas molecules. Increased sensitivity of rGO can be obtained by hybridization of this material with typical in gas sensor technology metal oxides (MOX), as SnO_2_ [3,4,5], WO_3_ [6,7], TiO_2_ [8,9], In_2_O_3_ [10], and ZnO [11,12]. One observes also hybridization of rGO with metal disulfides, as MoS_2_ [13,14,15] and WS_2_ [16], conducting polymers (graphene/PANI [17]) or with metal nanoparticles like Cu [18], Pt [19], Pd [20], and Pd-Pt [21]. It was also discovered that the working temperature of graphene/metal oxide sensors was lower than that of MOX sensors and sometimes can even reach room temperature (RT) [22,23,24,25]. Good sensitivity of GO to acetone at room temperature was obtained recently by using a microwave-based gas sensor realized as a coupled-line section covered with a graphene oxide film [26].

Another gas-sensitive oxide, α-Fe_2_O_3_ (hematite), is recently investigated as a NO_2_ sensor [27] but more frequently in the form of a nanocomposite with rGO [28,29,30]. The nanocomposites of α-Fe_2_O_3_/rGO are also used in detection of acetone [31,32], ethanol [33,34], and ethylene [35]. Among all crystal phases of iron oxide, essentially α-Fe_2_O_3_ is used in gas sensors technology. This phase is the most stable, environmentally friendly, sensitive, and easy in preparation. Mesoporous α-Fe_2_O_3_ samples with a large specific surface area were obtained with the use of templates [36,37]. Comparison of sensitivities for α-Fe_2_O_3_ and γ-Fe_2_O_3_ phases and their mixture is analyzed in [38].

Sensors of NO_2_ based on hematite, as can be inferred from the literature (e.g., see discussion in [27]), work at rather elevated temperatures of order 200 °C. In comparison to pure hematite, improved sensing properties vs. NO_2_ were obtained in hybrid nanocomposites rGO/Fe_2_O_3_ obtained by a facile hydrothermal method [28]. The enhanced response of the nanocomposite at RT was 3.86, compared to 1.38 for pure rGO. In [29] graphene-encapsulated α-Fe_2_O_3_ hybrids were investigated. The samples revealed good NO_2_ sensing properties at RT with the response time of order a few minutes. In this case, the hybrid was obtained by the hydrolysis of Fe^3+^ ions in the colloidal solution of GO and following hydrothermal procedure. The hydrothermal synthesis at 120 °C was applied by the authors of [30] to manufacture α

α-Fe_2_O_3_/rGO nanocomposites. In effect, nanosphere-like α-Fe_2_O_3_ of 40–50 nm diameters and single intercalated sheets of rGO were obtained.

In the present work, α-Fe_2_O_3_ and rGO were obtained separately and deposited on the substrate in the form of a multilayer by the spray method. Such a deposition method allowed for easy formation of slits and pores between the GO sheets, making diffusion of the ambient gas easier. Good sensitivity to NO_2_ was also obtained at temperatures close to RT but with quite high response and recovery times.

## 2. Materials and Methods

### 2.1. Sample Preparatiom

Graphene oxide (GO) was prepared using a wet chemical method from graphite (modified Hummers method). Obtained flakes consisted of 6–7 graphene layers with a total thickness of ca. 6 nm and reached the lateral size up to a few micrometers (manufacturing procedure of GO and rGO is described in detail in [39]). Ferric oxide α-Fe_2_O_3_ was also obtained by the wet chemical method as follows: to 100 mL of 0.1 M FeCl_3_, 300 mL of 0.1 M NH_4_OH solution was added. An obtained reddish brown suspension was heated for 60 min at 100 °C. The precipitate was washed with warm water on a filter with pore size of 0.2 µm. Both GO and Fe_2_O_3_ were deposited on the substrate by spraying, as is schematically illustrated in Figure 1.

The GO/Fe_2_O_3_ multilayer structure was deposited by LbL (layer by layer) coating technique using alumina substrate with interdigitated Au electrodes manufactured in thick film technology, Separation between electrodes was 0.3 mm. The geometry of electrodes with adequate masking enabled the deposition of two different structures with, e.g., different thicknesses. For investigations, two kinds of samples were prepared: 3 LbL structure consisting of 3 layers (Fe_2_O_3_-GO-Fe_2_O_3_) and 6 LbL structure (3 Fe_2_O_3_ and 3 GO layers) with Fe_2_O_3_ layer always on top.

### 2.2. Electrical Measurements

Measurements of sensor resistance were performed in conditions of selected temperature, humidity, and ambient atmosphere composition. The high-quality equipment (electrometer, current and voltage sources, scanners, mass flow controllers) of Agilent (Agilent Technologies, Santa Clara, CA, USA), Keithley (Keithley Instruments Inc., Cleveland, OH, USA) and MKS (MKS Instruments, Andover, MA, USA) manufacturers was used. The measurement chamber contained the sample, Pt 100 temperature probe and digital humidity sensor. The current was measured by an electrometer working in a constant voltage mode. Measurements of resistance varying in a range over nine orders of magnitude were possible. The temperature inside the chamber was changed by applying the voltage from an additional power source. All devices were controlled from the LabView custom application of Agilent Technologies interface card. The required NO_2_ gas concentration was obtained by controlling the ratio of gas to airflow rates. The humidity of the gas atmosphere was always kept constant with RH = 50% (detailed description of the measurement setup is given in [40]).

## 3. Results and Discussion

### 3.1. Structural and Morphological Characteristics

TEM measurements were performed with the help of high resolution scanning electron microscope HITACHI S-5500 (HITACHI Ltd., Tokyo, Japan) in a transmission mode. An example of a TEM image for the GO sample is shown in Figure 2a. Different transparencies for attenuated electron beams were caused by the stacking nanostructure of the investigated layers. Dark areas indicate the overlap of several layers. SEM pictures were obtained with the help of FEI Versa 3D Dual Beam microscope (Thermo Fisher Scientific, Waltham, MA, USA). In Figure 2b, one can see the flakes of GO sample deposited on Si/SiO_2_ substrate.

In the image of GO/Fe_2_O_3_ nanostructure, Figure 3a, one can see both gray areas of GO flakes and white spots of Fe_2_O_3_ nanoparticles of average dimension ca 100 nm, covering GO flakes. The XRD patterns, Figure 3b, were obtained by using an XRD diffractometer Bruker AXS D8 Advance Cr (Kα 2.2910 Å) (Bruker AXS GmbH, Karlsruhe, Germany) with vanadium filter 0.015 mm. Observed peaks belong to the crystal planes of the hematite phase, which can be indexed to the rhombohedral structure of α-Fe_2_O_3_ and their positions are in accordance with JCPDS 86-0550 card. Analysis of the X-ray reflectivity for GO/Fe_2_O_3_ nanostructure enabled the determination of its thickness, equal 47.5 nm, and the RMS of surface roughness, 1.42 nm.

### 3.2. Chemical Composition

The surface chemistry of investigated samples was analyzed using X-ray photoelectron spectroscopy. A Microlab 350 (Thermo Electron, VG Scientific) spectrometer with non-monochromatic Al Kα radiation (hν = 1486.6 eV, power 300 W, voltage 15 kV) was used for this purpose. The analyzed area was 2 × 5 mm. The hemispherical analyzer was used for collecting the high-resolution (HR) XPS spectra with the following parameters: pass energy 40 eV, energy step size 0.1 eV. The collected XPS spectra were fitted using the Avantage software (version 5.9911, Thermo Fisher Scientific), where a Smart function of background subtraction was used to obtain XPS signal intensity and an asymmetric Gaussian/Lorentzian mixed-function was applied. The position of the carbon C1s peak was assumed to be at 284.4 eV and used as an internal standard to determine the binding energy of other photoelectron peaks. XPS spectra for GO/Fe_2_O_3_ structure are shown in Figure 4 and for rGO/Fe_2_O_3_ structure in Figure 5.

A significant decrease of oxygen content in the samples after reduction is seen from the comparison of Figure 4 and Figure 5. The percent concentrations of oxygen-containing bonds C-O and C=O in both kinds of samples are collected in Table 1. In particular, the decrease in C-O bonds is clearly seen after thermal reduction (decrease from 34.2% to 20.5%). Similar observations are reported by other investigators of rGO/α-Fe_2_O_3_ composites, [28,32].

The recorded XR-XPS spectra of Fe2p revealed the chemical state of iron. Two wide peaks for both structures, located at the binding energies of ca. 711.0 eV and ca. 725.0 eV for Fe2p_3/2_ and Fe2p_1/2_, respectively, are characteristic for Fe^3+^ species in α-Fe_2_O_3_, which is in good agreement with other reports, e.g., [29,31].

### 3.3. Gas Sensing Properties

Measurements of resistance for GO/Fe_2_O_3_ multilayers in response to oxidizing NO_2_ atmosphere indicate that the structures behave as p-type semiconductors (under oxidizing atmosphere resistance of the sample decreases). Calculated gas response defined as S = (R_a_ − R_g_)/R_a_ where R_a_ and R_g_ are the sensor resistances in air and NO_2_, respectively, indicate that the structures are sensitive even at temperatures as low as 30 °C (Figure 6a,b). With increasing working temperature, the response also increased, as is shown for the structure GO/Fe_2_O_3_ 6 LbL with an increased number of layers, Figure 6c. As can be noticed, an increase in the number of layers in the structure does not influence the response remarkably. Variation of the baseline resistance with time at a constant NO_2_ concentration is shown in Figure 6d, and the comparison of sensitivities for the 6LbL multilayer and the single rGO layer is shown in Figure 6e.

The selectivity of GO/Fe_2_O_3_ 6LbL structure was tested by exposing the sensor to different interference gases, both oxidizing and reducing, as shown in Figure 7. Comparing the response to acetone (1.2% at 25 ppm) and hydrogen (0.9% at 50 ppm) with that to NO_2_, one can summarize that the influence of the tested interference gases is negligible. However, the influence of humidity on the sensor response is remarkable and is clearly seen at elevated temperatures. The example measurements of that response at 90 °C in a wide range of RH is shown in Figure 7b. The results indicate that both the increase and decrease of RH in comparison to 50% give variations of the response not higher than ±5%.

The response and recovery times of the structures are high, especially in the low operation temperature range. With increasing working temperature, these times significantly decreased, as shown in Figure 8 and Table 2.

### 3.4. Gas Sensing Mechanism

Fe_2_O_3_ is known as an n-type semiconductor, but the rGO/Fe_2_O_3_ hybrid structure behaves similarly to p-type rGO. Both chemisorbed O_2_ and NO_2_ act as electron traps decreasing the concentration of electrons. This is clearly seen in rGO in the decrease of resistance (hole density increases). Pure Fe_2_O_3_ is nearly insensitive to NO_2_, but in rGO/Fe_2_O_3_ composition, NO_2_ reacts with O_2_^−^ adsorbed on Fe_2_O_3_ surface, forming an intermediate NO_3_^−^ complex what was suggested in [30]
2NO_2 (gas)_ + O_2_^−^_(ads)_ + e^−^ → 2 NO_3_^−^_(ads)_

As a result, the unbalance of charge on the surface of Fe_2_O_3_ is compensated by transferring additional electrons from rGO to Fe_2_O_3_ surface, which results in additional holes in rGO, Figure 9, and then the increase in conductivity.

In Table 2, O_3_ on the increased sensitivity to NO_2_ can be understood from Figure 10. At the interface between rGO flakes and Fe_2_O_3_ grains, the p-n heterojunctions can be formed. The numerical value of work function for rGO was taken from [41], and the energy gap was evaluated from [42] for oxygen content in rGO equal O/C = 10%, as previously determined in elemental composition experiments [40]. The work function for Fe_2_O_3_ was taken from Guo et al. [31]. The difference in work functions of both materials causes the transfer of electrons from rGO to Fe_2_O_3_ and holes in the opposite direction. In effect, the hole accumulation region is formed on the p-rGO side. The concentration of holes in the accumulation layer on the p-side increases additionally after interaction with NO_2_, leading to the increased conductivity of GO flakes in the presence of NO_2_ gas, as schematically shown in Figure 11.

Potential barriers between Fe_2_O_3_ grains were not taken into account as the multilayer structure in response to NO_2_ behaves as a p-type semiconductor, then rGO layers play the essential role in conductivity.

As it was mentioned, the spray deposition technology allowed for easy formation of slits between the GO sheets, and additionally, Fe_2_O_3_ prevented these sheets from restacking. The slits can act as channels for gas diffusion, providing a higher number of active sites for the interaction with nitrogen dioxide, thus influencing the electrical transport.

## 4. Conclusions

Among air pollutants, NO_2_ is one of the most harmful gases, and measurements of its concentration with the help of inexpensive chemoresistive structures with very small power consumption are of great importance. In contrary to the pure Fe_2_O_3_ with small sensitivity to NO_2_ in the low-temperature range, rGO/Fe_2_O_3_ multilayer structures indicated good sensitivity to low concentrations of NO_2_ in the ambient air, going down to 1 ppm. Both materials of the structure were easily obtained by the wet chemical method and were sprayed on the substrate to form the needed composition. The sensitivity of the multilayer increased with temperature. Response time of order 1 min was possible to obtain at temperatures below 150 °C, but the recovery time was still high, of order tens of minutes. An increase in the number of layers of the structure did not influence the response remarkably. The sensing behavior of the multilayer structure was explained by both individual interactions of constituting materials with the ambient oxidizing gas and by the formation of heterojunctions at the contact regions of GO and Fe_2_O_3_.

## Figures and Tables

**Figure 1 sensors-21-01011-f001:**
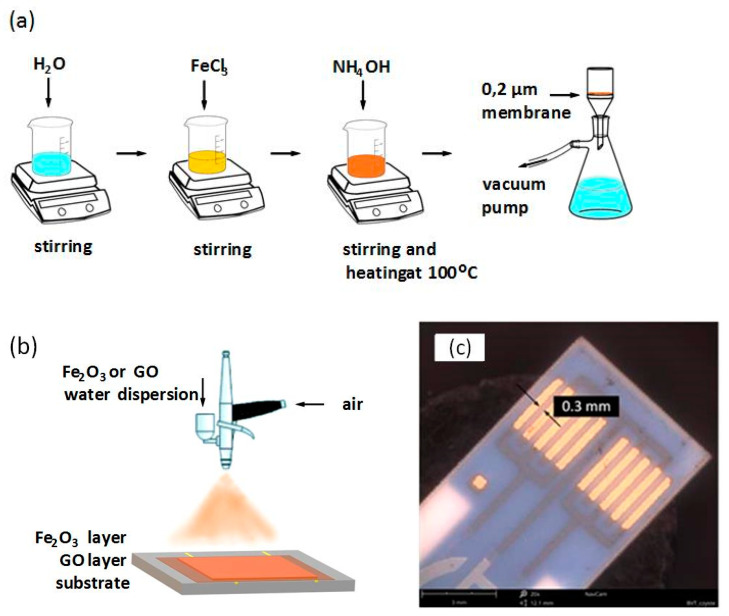
(**a**) Illustration of the manufacturing process of α-Fe_2_O_3_, (**b**) preparation of GO/α-Fe_2_O_3_ multilayer by the spraying method, and (**c**) outlook of the ceramic substrate used for deposition of the multilayer for electrical measurements.

**Figure 2 sensors-21-01011-f002:**
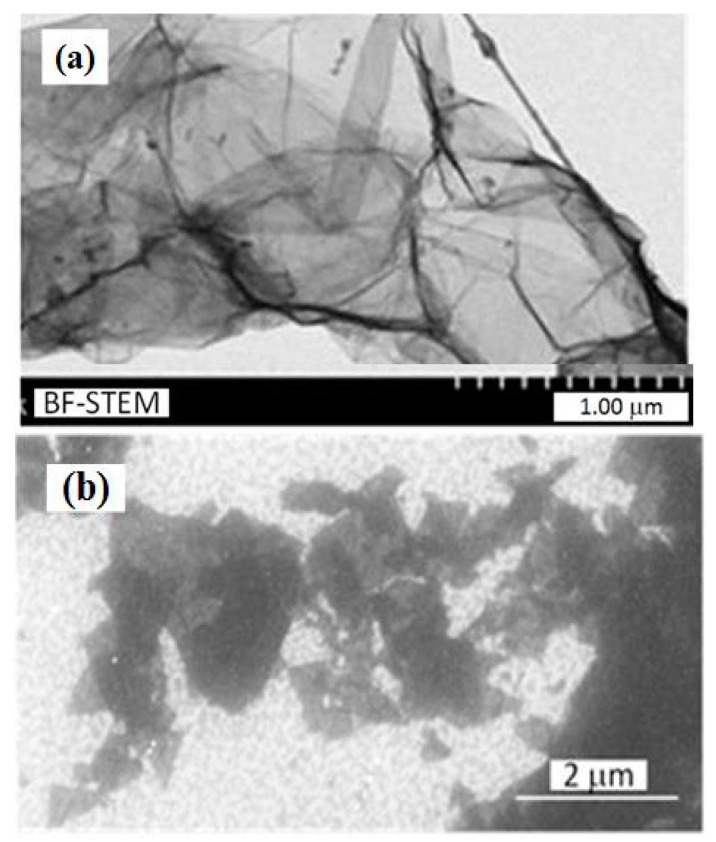
(**a**) TEM and (**b**) SEM image of GO flakes.

**Figure 3 sensors-21-01011-f003:**
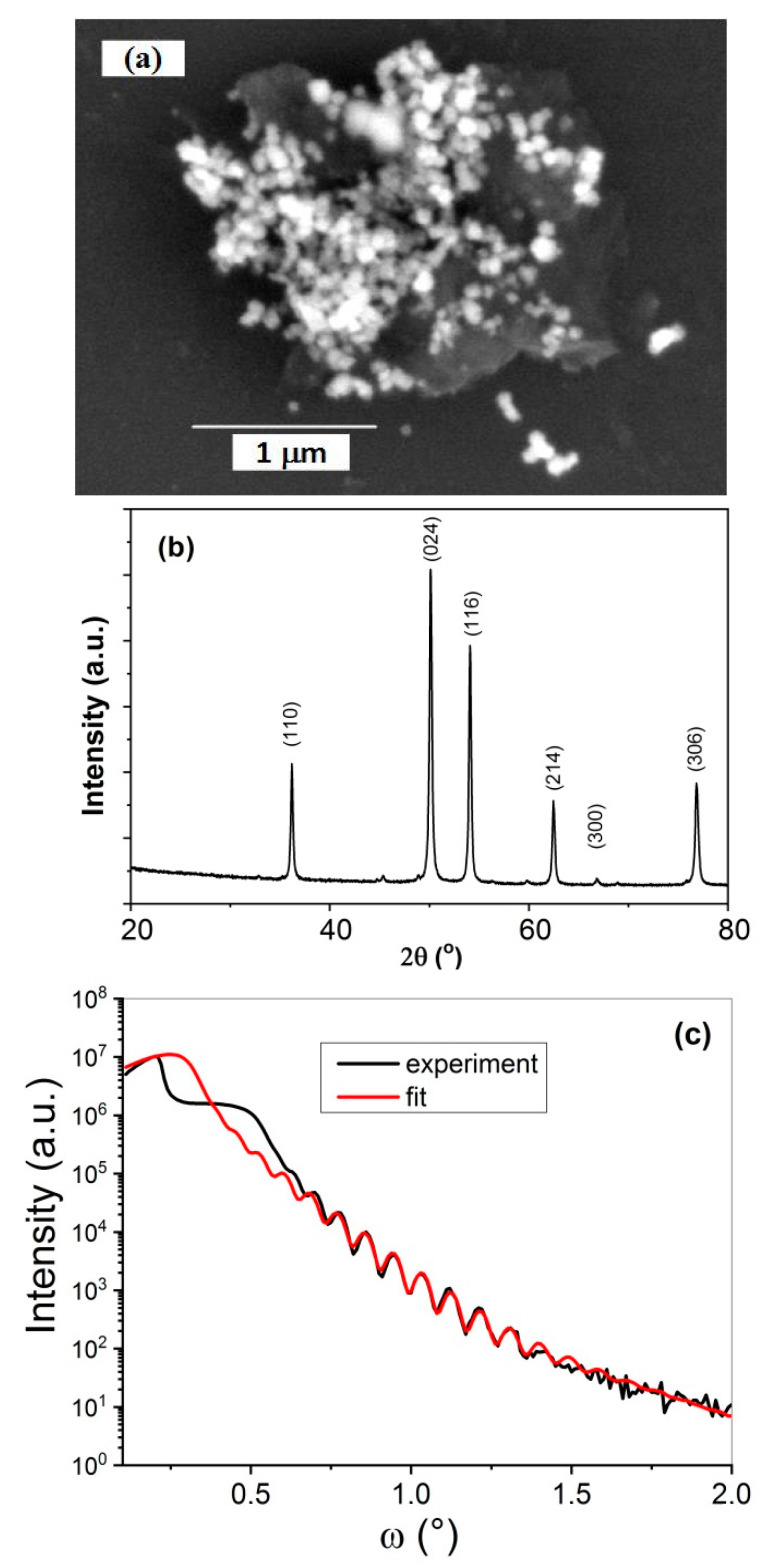
(**a**) SEM image of GO/α-Fe_2_O_3_ structure, (**b**) X-ray diffraction pattern for α-Fe_2_O_3_, (**c**) X-ray reflectivity pattern for GO/α-Fe_2_O_3_.

**Figure 4 sensors-21-01011-f004:**
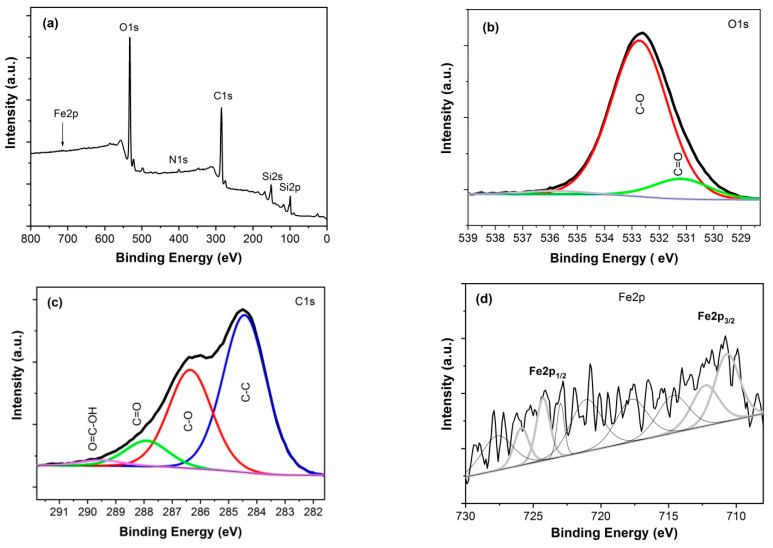
XPS spectra for GO/Fe_2_O_3_ multilayer: (**a**) wide scan, (**b**) HR O1s, (**c**) HR C1s, (**d**) HR Fe2p.

**Figure 5 sensors-21-01011-f005:**
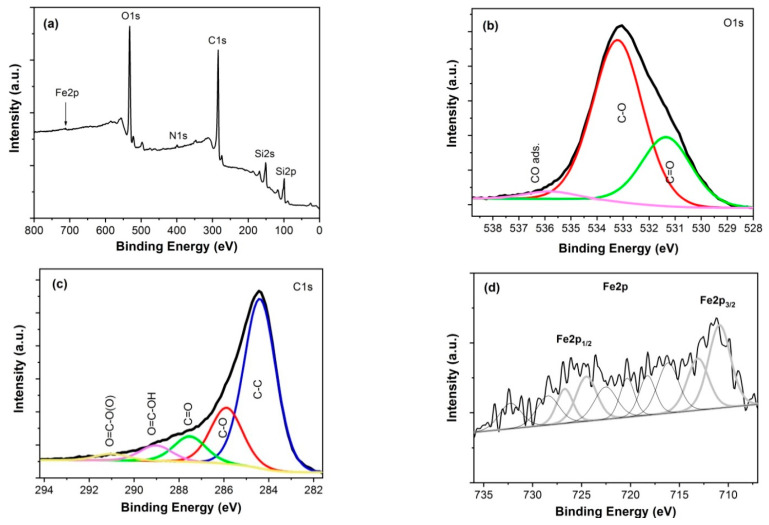
XPS spectra for rGO/Fe_2_O_3_ multilayer: (**a**) wide scan, (**b**) HR O1s, (**c**) HR C1s, (**d**) HR Fe2p.

**Figure 6 sensors-21-01011-f006:**
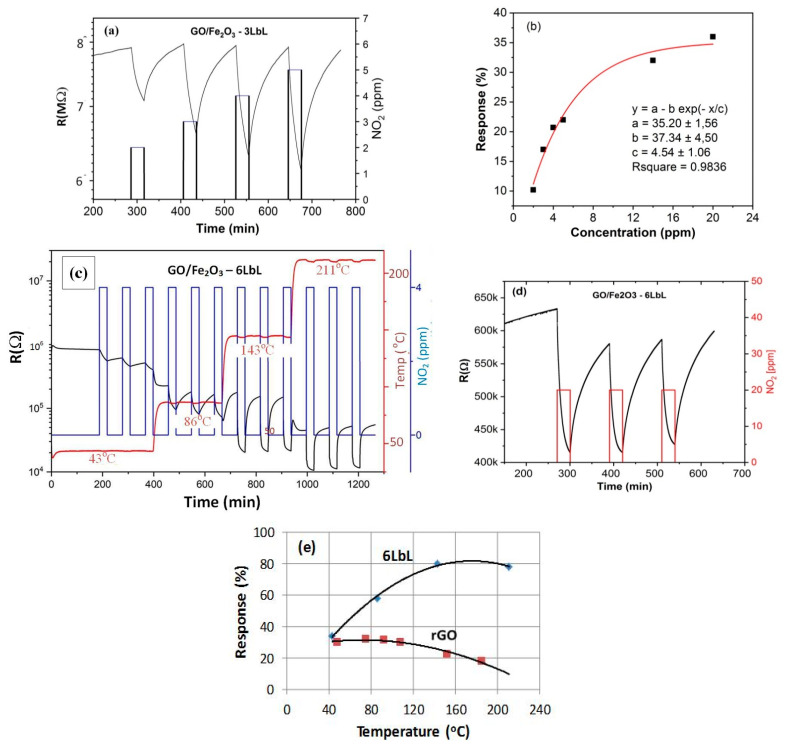
(**a**) Variation of resistance with time for GO/Fe_2_O_3_ 3LbL structure with increasing NO_2_ concentration at the temperature of 30 °C with a calculated response of this structure for increasing NO_2_ concentration (**b**); (**c**) variation of resistance for GO/Fe_2_O_3_ 6 LbL structure with changing sensor temperature at 4 ppm NO_2_ and (**d**) at 20 ppm NO_2_ at a constant temperature 30 °C; (**e**) comparison of responses for samples GO/Fe_2_O_3_ 6 LbL and single rGO layer with increasing measurement temperature at 4 ppm NO_2_.

**Figure 7 sensors-21-01011-f007:**
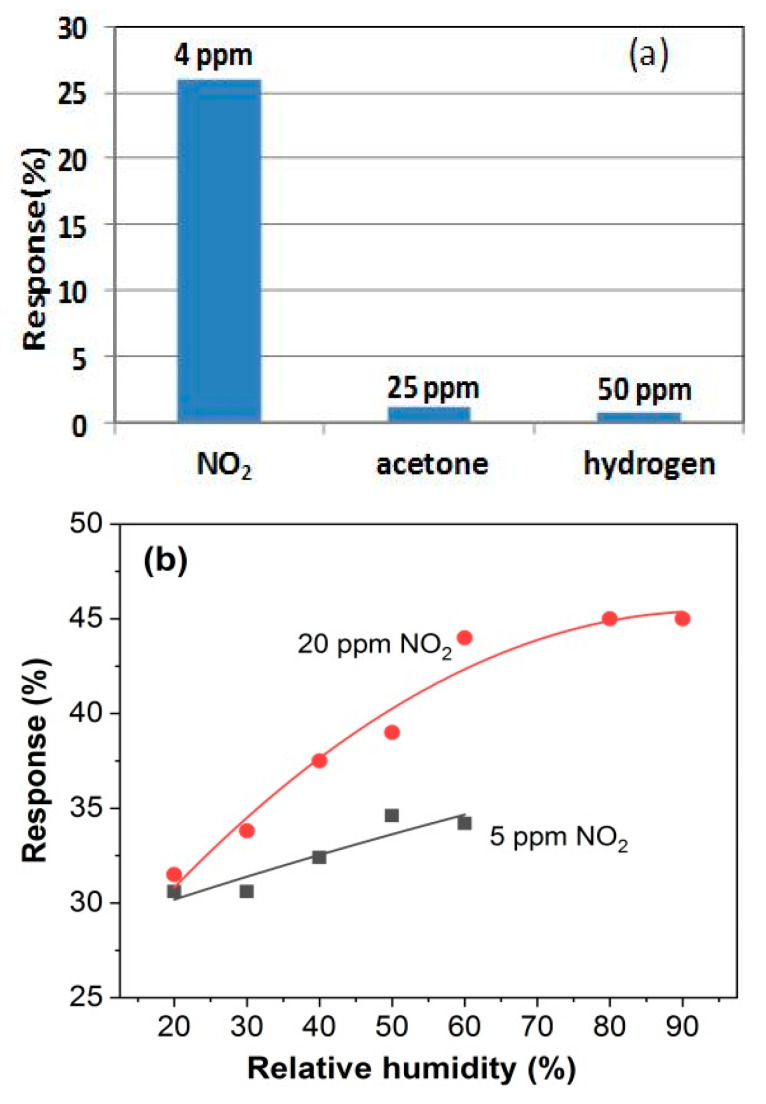
(**a**) The response of GO/Fe_2_O_3_ 6LbL structure to different gases at room temperature and (**b**) variation of this response with the change of relative humidity at 90 °C.

**Figure 8 sensors-21-01011-f008:**
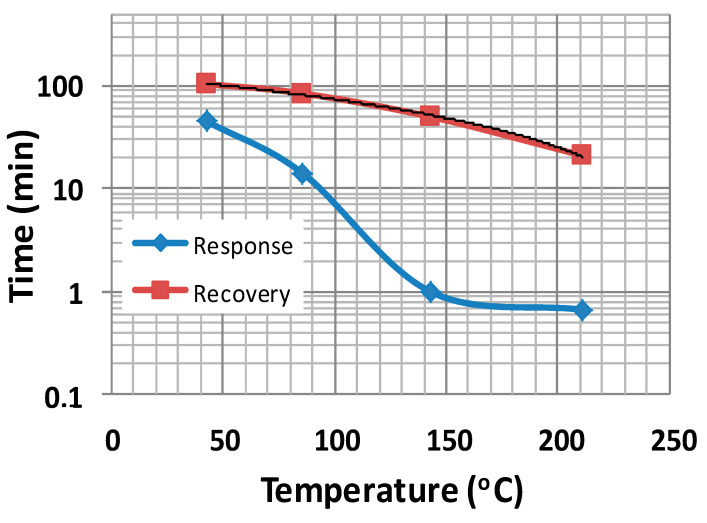
Response and recovery times vs. temperature for the sample shown in Figure 6c.

**Figure 9 sensors-21-01011-f009:**
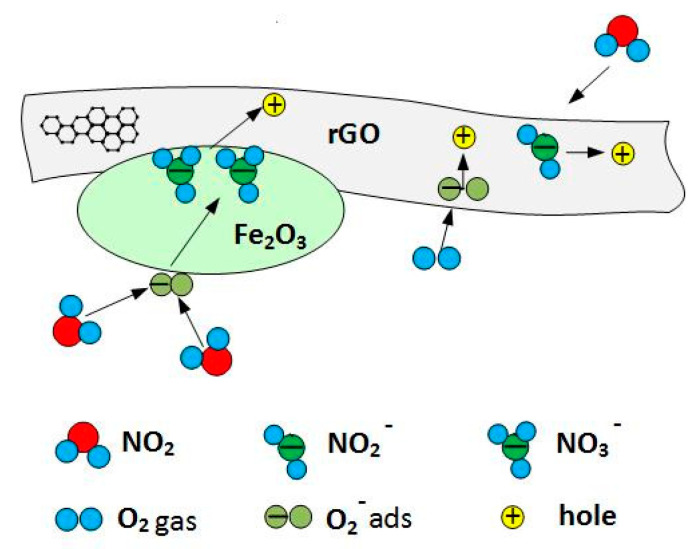
Interaction of NO_2_ gas with oxygen adsorbed on Fe_2_O_3_ surface can effectively increase the concentration of holes in rGO.

**Figure 10 sensors-21-01011-f010:**
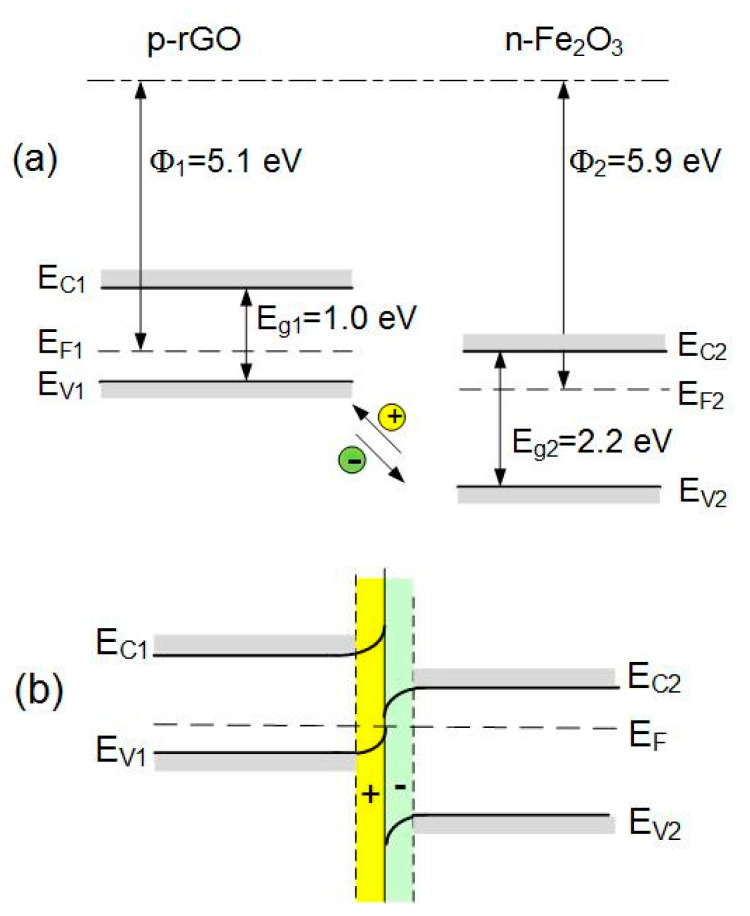
Diagram of energy bands of p-rGO and n-Fe_2_O_3_ before (**a**) and after (**b**) contact with the formation of a p-n heterojunction.

**Figure 11 sensors-21-01011-f011:**
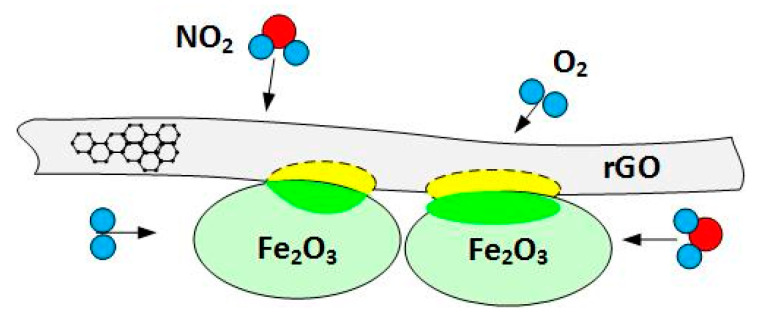
Interaction of Fe_2_O_3_ grains with rGO flakes leading to the formation of hole accumulation regions in rGO flakes.

**Table 1 sensors-21-01011-t001:** The percent contents of C1s chemical bonds in GO/F_2_O_3_ and rGO/Fe_2_O_3_ samples. The results are normalized to 100% for the C1s peak.

Sample	Chemical Bonds				
GO/Fe_2_O_3_	C-C (284.4 eV)	C-O (286.4 eV)	C=O (287.9 eV)	O=C-OH (289.5 eV)	-
	54.9%	34.2%	8.9%	2.0%	-
	**Chemical Bonds**				
rGO/Fe_2_O_3_	C-C (284.4 eV)	C-O (285.9 eV)	C=O (287.5 eV)	O=C-OH (289.0 eV)	O=C-O(O) (291.0 eV)
	62.2%	20.5%	9.3%	5.6%	2.4%

**Table 2 sensors-21-01011-t002:** Variation of response and recovery times with temperature for sample GO/Fe_2_O_3_ 6 LbL.

Temperature (°C)	Response Time t_res_ (min)	Recovery Time t_rec_ (min)	t_rec_/t_res_
43	45	104	2.31
86	14	84	6
143	1	50	50
211	0.67	21	31.3

## Data Availability

Data Availability on request.

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
