# Peer review of "Nitrogen Dioxide Sensing Using Multilayer Structure of Reduced Graphene Oxide and α-Fe2O3"

_sensors, 2021, doi:10.3390/s21031011_

Round 1

Reviewer 1 Report

The study is well done. The paper is easy to read and the sensor development is properly characterized.

To strengthen the paper

1) could the authors explain the advantage of using iron oxide in comparison to zinc oxide, since it has been shown the zinc oxide nanoparticles could achieve ppb level of sensitivity. 

2) one of the problems of oxide based sensors is the selectivity. Has the author performed selectivity studies against other oxide containing species such as carbon monoxide and carbon dioxide. Would the sensor be able to discriminate against them?

3) the authors should also provide more specifics on the device fabricated with emphasis on how the measurements are reliable. 

Author Response

Please find the responses in the attached file. 

Reviewer 2 Report

The letter "Reduced Graphene Oxide/Fe2O3 for Low Temperature Nitrogen Dioxide Sensing" is a poorly prepared scientific work. While just revising the two major sections of the manuscript, I do not find it suitable for publication. Thus I suggest the authors to work more on these sections

Abstract: I do not find any motivation statement in the abstract, what is the significance of the work? How is it different from any other work in the literature? 

Introduction: Most of the statements in the first 2 paragraphs of the introduction requires citation. This section is very poorly written. The authors failed to provide good insights/background of this work in relevance to the literature. English/grammar is also very poor.  

My suggestions for the authors (mainly the first authors) is that you should first have clear understanding of the field of your research. Whats going on? what is the trend? What is the limitation? what are the advancements? Then you need to plan your work based on the need/importance? For me, there is no novelty factor in your work nor any significance in the field.

Author Response

(The authors gave the same response as above.)

Reviewer 3 Report

The submitted manuscript “Reduced graphene oxide/Fe2O3 for low temperature nitrogen dioxide sensing” presents experimental results of the sensing behavior of a sensor fabricated via a layer by layer spraying technique. Analysis of acetone and hydrogen interference was also included, as well as a proposal of the gas sensing mechanism. The importance of the study is high in the field of NO2 sensors, however a major revision is recommended based on the following questions and comments.

  1. The title of the manuscript does not indicate the type of structure (multilayer) that was fabricated for the proposed sensor and that is different from other structures already reported in literature.
  2. In the introduction, the importance of sensing nitrogen dioxide in the motor vehicles exhaust was mentioned, however later on, when the results indicate that response (%) increases with temperature for a certain range, there is no mention at all on the effects for a possible application. It would be good to discuss results in regard of applications.
  3. There is no mention about the reproducibility of the samples prepared by the proposed method. What is the largest sample that can be prepared by this technique? How uniform, regarding thickness and porosity, are the obtained samples? I think all this is crucial.
  4. Have you evaluated degradation/life time of the sensors? How is it compared to bench mark NO2 sensors?
  5. Regarding your results, it would be good to compare them with those from literature, and include a mention in the discussion, as to emphasize the pros and cons (development opportunities) of the multilayered reduced graphene oxide/ Fe2O3.
  6. If possible, include uncertainties (do the error/uncertainty propagation) of all numerical results.
  7. Samples were fabricated using two substrates (Si/SiO2 and alumina), however I did not find results of the effect of the substrate on the structure and the sensing behavior. And there is no indication of what substrate was used for each reported sample.
  8. Details of the complete sensor fabrication are missing in the Materials and Methods section. What type of electrodes were used and how were they prepared?
  9. Why was relative humidity set to 50 % for electrical measurements?
  10. There is no description of the system for electrical measurements regarding heating and temperature sensing.
  11. Have you performed the sample characterization after measurements?
  12. Regarding figure 6a), what was the explanation for the behavior of the presented curve before the first NO2 pulse (0-140 min)?
  13. What would be a possible explanation of the behavior of curves shown in figure 6 d)?
  14. About the proposed gas sensing mechanism, would it be possible to track changes in the oxidation states of iron as an evidence of the mechanism?
  15. Some typographical errors were found and marked in the attached file.

Author Response

(The authors gave the same response as above.)

Reviewer 4 Report

I recommend major revisions on the specific publication before recommending it for publication.

1.) The article introduction does not discuss other types of other sensors and sensing techniques commercially used, Such as LSM, YSZ, etc.

https://doi.org/10.1016/j.snb.2017.10.014

https://www.jstor.org/stable/44730751

https://doi.org/10.1016/j.snb.2010.01.036

This particular sensor's specific advantage would have compared to other types like impedance metric, potentiostatic, etc. Need to discuss that too?

2.) Table the numbers are in Atomic %! Mention it. The authors mention there is a significant reduction in the C-O bond after thermal reduction. Can they explain how does it affect that particular sensor performance?

3.) Besides rGO and GO with Fe2O3, there are no other types of sensors compared. It is challenging to compare how better performing this particular sensor is? 

4.) No demonstration of how the 50% humidity affects sensor performance (dry and wet data).

5.) No sensitivity quantification is performed via graph (Signals Vs. range of NOx in ppm).

6.) What are the LOD (limits of detection) and LOQ (Limits of Quantization)?

7.) Interferents like CH4, CO, CO2 needed to be studied and how it is affecting sensors performance.

8.) What is the recovery rate of the sensor? No tabulation is provided.

9.) The conclusion is not well written. It should be more quantifiable in terms of the findings that the authors have.

Author Response

(The authors gave the same response as above.)

Round 2

Reviewer 2 Report

Sensors and Actuators B 241 (2017) 109–115

Sens. Actuators B 2018, 261, 252-263

RSC Adv. 2014, 4, 57493-57500

Please see these papers which are also using same composite material and for same application but much better than your letter. I still do not find this letter attractive for publication, just a different fabrication method doesn't warrant publication in Sensors 

Author Response

Dear Reviewer, please find our responses attached. 

Reviewer 3 Report

Most of my previous comments were taken into account. I recommend a minor revision based on those that were not fully considered:

  • What is the largest sample that can be prepared by this technique? I ment what is the size of it?
  • I previously recommended to include uncertainties (do the error/uncertainty propagation) of all numerical results. Those of the chemical anaylisis results are still missing.
  • During the first revision process, some typographical errors were indicated with comments and marks in the original PDF file, however not all of them were corrected. I expected authors would more carefully revise the manuscript (see attached file).

Author Response

(The authors gave the same response as above.)

Reviewer 4 Report

I recommend it to get published.

Author Response

Dear Reviewer, thank you very much for your help with improving our paper.